# Novel Deep-Learning Modulation Recognition Algorithm Using 2D Histograms over Wireless Communications Channels

**DOI:** 10.3390/mi13091533

**Published:** 2022-09-17

**Authors:** Amr Marey, Mohamed Marey, Hala Mostafa

**Affiliations:** 1Department of Electrical and Computer Engineering, Faculty of Engineering, University of Alberta, Edmonton, AB T6G 1H9, Canada; 2Smart Systems Engineering Laboratory, College of Engineering, Prince Sultan University, Riyadh 11586, Saudi Arabia; 3Department of Information Technology, College of Computer and Information Sciences, Princess Nourah bint Abdulrahman University, P.O. Box 84428, Riyadh 11671, Saudi Arabia

**Keywords:** modulation recognition, deep learning, convolutional neural network, 2D in-phase quadrature histogram

## Abstract

Modulation recognition (MR) has become an essential topic in today’s wireless communications systems. Recently, convolutional neural networks (CNNs) have been employed as a potent tool for MR because of their ability to minimize the feature’s susceptibility to its surroundings and reduce the need for human feature extraction and evaluation. In particular, these investigations rely on the unrealistic assumption that the channel coefficient is typically one. This motivates us to overcome the previous constraint by providing a novel MR suited to fading wireless channels. This paper proposes a novel MR algorithm that is capable of recognizing a broad variety of modulation types, including *M*-ary QAM and *M*-ary PSK, without enforcing any restrictions on the modulation size, *M*. The analysis has shown that each modulation choice has a distinct two-dimensional in-phase quadrature histogram. This property is beneficially utilized to design a convolutional neural-network-based MR algorithm. When compared to the existing techniques, Monte Carlo simulations demonstrated the success of the proposed design.

## 1. Introduction

Research into the investigation of communication signals from unspecified or partly recognized sources, with the aim of finding the transmission characteristics peculiar to the utilized broadcaster, has grown in popularity with the development of wireless communication systems. The extent of this activity, which is known as signal recognition, spans from recognizing signal specifications such as modulation format, coding scheme, antenna setup, and data rate [1,2,3,4,5]. Historically, the military has been the primary user of signal-recognition technologies. This is because the detection, analysis, and recognition of unknown signals from possibly hateful communication sites are essential tasks in signal interception, radio monitoring, jamming interference recognition and reduction, and electronic warfare [6]. Signal recognition is still an important part of modern sophisticated radios used for defense communications. The advent of software defined radios, versatile transceivers that can modify transmission characteristics such as modulation and coding formats, has sparked a renewed interest in signal-recognition systems within the framework of commercial communications [7,8]. These characteristics may be shared across software defined transceivers through specific routes; however, doing so wastes communication bandwidth and slows down the transmission rate. To avoid this, one solution is to use signal-recognition techniques to extract these characteristics from the heard signal on the receiving site.

As the number of customers and the desire for network services has grown in private wireless communications over the last few decades, the wireless spectrum has become more congested. This makes it hard for wireless communication developers to increase capacity while keeping energy use low and minimizing interference. This issue is being addressed by the growing cognitive radio approach, which can be seen as the next step in the development of the software-defined radio idea [9,10]. This methodology makes it possible to manage radio resources in a way that is both dynamic and demand-driven. The goal of cognitive radio systems is to optimize the use of existing communication facilities by adapting transmission settings to the specific characteristics of the wireless frequency surroundings. One of the main distinctions between traditional and cognitive transceivers is that the latter must be conscious of the suppliers and transmission settings in their frequency surroundings. This means that cognitive radio is one of the most cutting-edge and exciting uses for signal-recognition systems in the commercial sector.

Modulation recognition (MR) is an essential element of signal-recognition systems that predicts the most likely modulation type for an observed signal based on its structure [11,12]. At each frame, the transmitter regulates the data rate demand while maintaining a particular level of service by picking a modulation scheme from a pool of modulation possibilities. Modulation information may be present in each frame to ensure that the recipient is aware of the modulation choice and can respond appropriately. However, this tactic is inadequate in most practical applications because the additional information reduces the spectrum efficiency. As a consequence, MR is utilized at the receiver to identify the modulation type of incoming frames, avoiding the unnecessary overhead. Eventually, the frames are appropriately decoded, and the information symbols are precisely retrieved.

Most available MR solutions in the literature are generally designed using one of two strategies: likelihood-based (LB) and feature-based (FB) [13]. The former computes the probability functions of all possible modulation schemes for the received signal and selects the one with the best likelihood. In theory, LB algorithms offer the best recognition performance, but they have a high computing cost and lack of resilience against transmission mismatch. These constraints make LB MR implementation unfeasible. On the opposite side, because of their reduced computing complexity, FB algorithms are recognized as a viable alternative to LB techniques. The use of artificial intelligence methods in MR has been more popular during the last several years [14,15]. It is worth mentioning that the implementations of artificial intelligence approaches have been progressively developing in a variety of technical disciplines [16,17,18,19,20,21,22].

The studies reported in [23,24] assume that the channel coefficient is always one, which is not a reasonable constraint in practice. The primary contributions of this study are the following.
We loosen the prior limitation by proposing a novel MR applicable to flat-fading wireless channels.We show that each modulation choice has a distinct two-dimensional in-phase quadrature histogram (2-D IQH), which is beneficially utilized to design a CNN-based MR algorithm.In addition to operating over practical wireless environments, the proposed algorithm provides much less complexity when compared to [23,24]. It only requires two CNN deep layers, which greatly shortens the training and recognition times. The conceptual diagram of the proposed receiver is shown in Figure 1.

The rest of this work is organized in the following manner. Related works are discussed in Section 2. A description of the system model is introduced in Section 3. Data generation and feature extraction are described in Section 4. The proposed MR algorithm is provided in Section 5. The results of Monte Carlo simulations are analyzed in Section 6. Finally, the work is closed in Section 7.

## 2. Related Works

A recently popular work that applies two separate FB algorithms for MR is [25]. Both algorithms utilize the cumulative distribution function of the amplitudes of the received signals as a feature and consider their classification pool to be 4-QAM, 16-QAM, and 64-QAM. The first algorithm applies a tuned support vector machine (SVM) for MR, and the second algorithm utilizes a neural network (NN) as its classifying mechanism. The results for both algorithms show high MR performance for an SNR above 16 dB. While these results seem tolerable, there are some hidden downfalls. First, the algorithms are only applicable at high SNRs, which is far from optimal in many real-world situations. The findings of the investigation that is described in [26] led to the same conclusions as well. Furthermore, the classification pool is limited to QAM constellations. Since the symbol amplitudes in any PSK constellation are identical, the classification performance of the two classifiers would suffer greatly if there were more than one PSK constellation in the pool. Because of this, additional attributes, such as the phase of the received symbols, are needed to properly categorize the PSK constellations. In this work, we simulate the methods provided in [25], expanding the modulation pool to include several PSK constellations in order to demonstrate the algorithms’ shortcomings.

When compared to traditional statistical and hypothesis testing, deep learning’s recognition accuracy is far superior, and its capacity to learn directly from high-dimensional raw input data surpasses that of machine learning. Convolutional neural networks (CNNs), which utilize an FB approach, have recently attracted particular attention for MR challenges. This is due to their characteristics of reducing the necessity for human feature extraction and assessment, as well as minimizing the feature’s sensitivity to the surroundings [14]. P. Peng et al. have proposed a CNN-based MR algorithm utilizing image processing techniques [23]. The received samples are converted into a three-channel RGB image format, with each modulation format exhibiting a distinct RGB image representation. This image was fed into the predefined AlexNet [27] and GoogLeNet [28] CNN models to conduct MR.

Y. Kumar et al. created a transfer learning-based MR technique that utilizes the ResNet-50 and Inception ResNet V2 CNN models [24]. To produce color images, the constellation density of the received signals was processed through various filters comprising three masks. Training and recognition periods for this algorithm are very lengthy since the former CNN models are 50 layers deep while the latter one requires 164 layers.

## 3. Preliminary Studies

We explore two-dimensional histograms of various modulation types at the receiver in order to expose differentiating characteristics that may be used to categorize them. The following common assumptions guide the investigation.

The transmitter broadcasts uncorrelated information symbols d(k) with
(1)$$E\left[d\left(k\right)d^{*}\left(k^{\prime }\right)\right]=\left\{\begin{array}{cc} \sigma_{s}^{2} & \mathrm{if}\;k=k^{\prime }\\ 0 & \mathrm{otherwise} \end{array}\right.$$
where d(k) is the *k*th transmit symbol belonging to a constellation θ of order *M*, $E\left[\cdot \right]$ represents the statistical expectation, and * is the complex conjugate operation. Without sacrificing generality, we set $E\left[{\left|d(k)\right|}^{2}\right]$ to the value of 1. Here, the transmitter picks the constellation θ from a list of available options based on the strength of the channel.The channel gain h(k) is modelled as a zero-mean complex Gaussian random variable, with $E\left[{\left|h(k)\right|}^{2}\right]=1$.Noise samples ω(k) are considered to be Additive White zero-mean Gaussian Noise (AWGN) samples that are symmetric, independent, and identically distributed, with variance $\sigma_{n}^{2}$.

As such, the recieved signal x(k) can be modeled as
(2)x(k)=h(k)d(k)+ω(k)

Before the signal is applied to the proposed CNN, the channel is first estimated through a generic channel estimation algorithm. Some recent advances in channel estimation include [29,30]. The estimated channel coefficient is said to be
(3)$$\hat{h}\left(k\right)=h\left(k\right)+\Delta h\left(k\right)$$

The estimation error resulting from the channel estimator (CE) can be taken into account through Δh(k). After h(k) has been estimated, x(k) is transformed to r(k) through the following equation:(4)$$r\left(k\right)=\frac{x(k)}{\hat{h}\left(k\right)}=\frac{h(k)}{\hat{h}\left(k\right)}d\left(k\right)+\frac{\omega (k)}{\hat{h}\left(k\right)}$$

If $\hat{h}\left(k\right)⋍h\left(k\right)$ and the CE has low estimation error variance, then
(5)$$r\left(k\right)\approx d\left(k\right)+\frac{\omega (k)}{\hat{h}\left(k\right)}$$

As such, the multiplicative effect of the channel has been eliminated. This shows that the 2-D IQH of the recieved signal x(k) can be used to extract d(k) through a well-designed CNN.

Figure 2 shows the two-dimensional histograms of QPSK, 8-PSK, 16-QAM, and 64-QAM received signals at a signal-to-noise ratio (SNR) of 12 dB. The *x*-axis and *y*-axis represent the in-phase and quadrature components of the received signal, respectively, and the *z*-axis refers to the number of occurrences of each (*x*,*y*) combination. Here, both the *x*-axis and *y*-axis are divided into 80 distinct levels leading to 80×80 blocks in the xy plane. Each block’s height specifies the number of received samples that correspond to that block. As observed from the figure, histograms differ from one modulation type to another. We leverage this tendency as a property to design a novel MR algorithm relying on CNN architecture.

## 4. Data Generation and Feature Extraction

In this section, we demonstrate how to produce the data-set and then convert it to a format that is suitable for being processed by a CNN network. We use numerical values rather than symbols to simplify the explanation. However, alternative values can be easily used as appropriate for the particular application. We set the SNR to have a range that goes from −6 dB to 15 dB, with increases of 3 dB at each step. The modulation options under consideration are BPSK, QPSK, 8PSK, 16QAM, 32QAM, and 64QAM. Using (Equation 2), we create 200 received signals for each possible value of SNR and each available modulation option. Each signal has 8192 samples, and the total number of signals is 9600. The created data are laid out in the format displayed in Table 1.

After that, the received signal is transformed into a matrix of two dimensions, as shown in the following steps, so that it can be analyzed by a CNN network.
We split the incoming signal into its in-phase and quadrature components, $I_{r}$ and $Q_{r}$, respectively.We determine the maximum and minimum values for each of $I_{r}$ and $Q_{r}$.We create 80 bins across the entire scale of $I_{r}$ and $Q_{r}$.We calculate the number of samples that fit inside each grid. This matrix represents the histogram of the received signal. The prior information is visualized in Figure 3.

It has been discovered that there is a plateau in correct recognition rate improvements at the number of bins of 80. Here, the employed criterion is that the recognition rate’s accuracy cannot fluctuate by more than 0.5 percent from the previous value. Using more complex optimization techniques to ascertain the requisite number of bins is useless because they will increase the system’s complexity and run time without providing much benefit.

## 5. Proposed MR Algorithm

### 5.1. CNNs Architecture

CNNs are feed-forward neural networks that employ a 2D convolution operation rather than matrix multiplication [14]. CNNs have a three-layer structure consisting of a convolutional layer, a pooling layer, and a fully connected layer. The convolutional layer is the critical component of CNNs that carries out the majority of the computations. This layer consists of several 2D convolutional filters, which have a size between 1×1 and 4×4. The transfer function of a convolutional layer is
(6)$$y_{n}=g\left(a_{n}+\sum_{m}k_{mn}⋆x_{m}\right)$$
where $y_{n}$ is the output feature map regarding the *n*th filter, $x_{m}$ is the *m*th input activation, $a_{n}$ represents the learned bias, ⋆ denotes the 2D convolution operation, and $k_{mn}$ represents the filter weight parameter. Here, g(.) represents the non-linear activation function [31].

By making the kernel size smaller than the input size, CNNs can achieve sparse connectivity between input and output units, which enhances statistical efficiency and reduces memory needs. Furthermore, there is a possibility that numerous neurons will make use of the same parameter multiple times. As a result, the model’s demands with regard to storage space can be greatly cut down.

The pooling layer is responsible for replacing the output at specific locations with an appropriate statistic that is compiled from the outputs of surrounding locations. This layer consists of sliding a 2D filter over various sections of the feature map in order to minimize the feature map’s dimensionality. Popular pooling layers regularly used in CNNs are max-pooling layers and average-pooling layers. Max-pooling layers select the element with the maximum value. Thus, the most salient characteristics from the original feature map would be retained in the resulting reduced feature map. Average-pooling layers calculate the average value of the the elements in specific regions of the feature map. Therefore, the reduced feature map would be made up of the mean values of those regions.

The fully-connected layer is identical to the layers of artificial neural networks, where each neuron is connected to all the previously activated neurons from the previous layer [32]. A general illustration of the convolutional layer, the pooling layer, and the fully connected layer in a CNN is shown in Figure 4.

It is also common for CNNs to have a batch normalization layer that serves to standardize the underlying feature map. There is also usually a rectification layer (ReLU) layer that takes the absolute value of each element *x* in its input feature map; this can be described as [14]
(7)$$\mathrm{ReLU}\left(x\right)=\left\{\begin{array}{cc} x & x>0\\ 0 & x\leq 0 \end{array}\right.$$

CNNs applied to multi-class classification problems utilize the cross entropy loss function in order to optimize the model. The cross entropy function is defined as
(8)$$L=-\sum_{m=1}^{\alpha }\sum_{n=1}^{\beta }b_{mn}\ln\left(c_{mn}\right)$$
where α is the number of training examples, β is the number of classes present, and $b_{mn}$ denotes the actual class of the *m*th training sample associated with the *n*th class. $c_{mn}$ represents the output predicted by the network for the the *m*th training sample associated with the *n*th class [14].

In general, in order to apply CNNs to the MR problem, three steps are applied. First the CNN network architecture is designed. This is shown in Section 5.2. Next, the CNN trains itself by minimizing the loss function L. Finally, the CNN model is validated and tested to confirm its applicability.

### 5.2. Modulation Recognition CNN Structure

The proposed MR CNN algorithm exploits the 2D IQH feature of the modulated received signal. After the 2D IQH of the received signal is extracted, it is inputted into the proposed CNN. The loss function used to train this CNN is the cross entropy function L as shown in (Equation 8).

The designed CNN is presented in Figure 5, and its hyper parameters are presented in Table 2. The input image layer is designed to accept 80×80 matrices. The purpose of the input layer is to simply receive the 2D IQH of the received signal and output it to the first convolutional layer (CL I). The task of the convolutional layer is to extract the high level features of the 2D IQH through multiple kernels through the 2-D convolution operation mentioned in (Equation 6). The received matrix of the convolutional layer is padded with two layers of zeroes, and there are 128 filters such that each filter has a kernel size of 3×3. The outputs of CL I are passed on to the batch normalization layer and the ReLU layer. The former is responsible for normalizing the convoluted matrix to minimizing the internal co-variate shift and the latter serves as an non-linear activation function [15]. The next layer of the network is the max-pooling layer [15]. This pooling layer has a pool size of 2×2 and a stride of 1. The next layers of the CNN consist of another convolutional layer (CL II), batch normalization layer, reLU layer, and max-pooling layers; each of these layers has identical parameters to the former layers. After the two deep layers are generated, the final stage of the CNN consists of the artificial neural network, whose task is to classify the modulation type of the received signal based on the feature extractions of the earlier stages of the CNN.

## 6. Simulation Work

### 6.1. Experimental Environment

Extensive Monte Carlo simulations have been carried out in order to evaluate how realistically the proposed technique performs. For each frame of transmission, the source chooses a modulation scheme from a pool of six options based on the quality of service and capacity requirements. The potential candidates are BPSK, QPSK, 8-QAM, 16-QAM, 32-QAM, and 64-QAM. These modulation formats have been used in many wireless standards, including fourth and fifth cellular system generations, wireless local area networks, and satellite systems [7,8]. It is essential to keep in mind that the proposed strategy is generic in the sense that it can be utilized for any number of different modulation schemes. We produce 9600 distinct received signals for each modulation choice at various SNR values. Each signal is represented by 8192 samples. The channel h(n) and noise ω(n) are assumed to be derived from a complex-valued Gaussian distribution to model real-life channel and noise characteristics. The mean power of the transmitted signal is equal to one, i.e., $E[{\left|x(n)\right|}^{2}]=1$. Simulations are run with SNRs ranging from −6 dB to −15 dB in 3 dB increments. The simulated received signal data set is described in full by the above given details. The simulations were carried out on a personal computer with Matlab software and the settings listed in Table 3. The training time was 5.6 s while the testing duration was 2.4 ms.

### 6.2. Results and Discussion

Figure 6 shows the correct recognition rate as a function of SNR for each possible modulation option. The proposed algorithm fails to deliver reliable recognition results below an SNR of 3 dB. This is due to the fact that the noise is predominant, which makes it difficult for the proposed algorithm to function well. The proposed algorithm, on the other hand, is able to obtain nearly flawless recognition rates for all of the modulation schemes that are being taken into consideration when the SNR is more than 5 dB. Consequently, this proves that the proposed CNN structure is successful.

To enable comparison, Figure 7 displays the average recognition performance of the proposed algorithm and that which can be achieved by employing the methods detailed in [25]. The average recognition performance is defined as $\frac{1}{6}\sum_{\lambda \in \Theta }\mathrm{Pr}\left(\lambda |\lambda \right)$, where $\mathrm{Pr}\left(\lambda |\lambda \right)$ is the probability that the recognized modulation is λ when the modulation used by the transmitter is λ. The summing is executed for all λ values in the set Θ= BPSK, QPSK, 8PSK, 16QAM, 32QAM, and 64QAM. Note that we rely on the average recognition rate to decrease the number of curves presented in the figure; instead of displaying 18 curves, we just show three curves. This helps to make the figure more readable and understandable. It is not difficult to see that the proposed algorithm is vastly superior to the algorithms described in [25]. This is due to the fact that both algorithms utilize the cumulative distribution function of the received signal amplitude. All M-PSK modulation forms maintain the same amplitude, making it impossible for the aforementioned methods to distinguish between them.

It was assumed throughout the previous discussions that the receiver’s estimation of the channel coefficient was accurate. To take a level further in practical applications, the subsequent step is to examine at how vulnerable the suggested recognition method is to variations in channel estimation accuracy. Figure 8 depicts the average recognition rate as a function of channel estimation error variance, $\sigma_{h}^{2}$, at various SNR setting. The results show that the proposed technique is robust to the channel estimation error and provides appropriate performance throughout a wide range of $\sigma_{h}^{2}$. The previous fact is supported by the confusion matrices for SNR = 12 dB and $\sigma_{h}^{2}$ = 0.0001, 0.01, and 0.1, as illustrated in Table 4, Table 5 and Table 6, respectively. Additionally, Figure 9 shows the false acceptance rate as a function of the variance of channel errors at SNR = 12 dB. Here, the false acceptance rate for a certain modulation β is defined as $\sum_{\lambda \in \Theta ,\lambda \neq \beta }\mathrm{Pr}\left(\beta |\lambda \right)$. The results show that for all modulation schemes, the percentage of false acceptance rises with the variance of channel errors and is essentially independent of the kind of modulation.

Since cognitive receivers usually work in an environment with interference, we evaluate the impact of digitally modulated narrow-band interference signals on the performance of the proposed algorithm. Figure 10 displays the average recognition performance as a function of signal-to-interference ratio (SIR) across a variety of SNRs. The interfering signals are generated in the manner outlined in [33]. When both SNR and SIR are higher than 9 dB, the results show that the suggested algorithm is able to achieve nearly perfect recognition performance. Even with high SIR values, the proposed algorithm has a poor recognition performance at low SNR levels. This is because the background noise is so pervasive.

## 7. Conclusions

This paper proposes a novel algorithm for modulation classification (MR) in wireless communications systems. The proposed approach is flexible in the sense that it can be used to a wide range of modulation schemes. According to the theoretical analysis, each modulation option has a distinct two-dimension histogram. This is utilized as a characterized feature by a convolutional neural network (CNN). The CNN is then configured to produce optimal MR results at the minimum SNR possible. The proposed structure for the CNN consists of only two layers, which drastically cuts down on the complexity of the processing. When compared to prior works, it is shown that the suggested algorithm yields superior MR rates.

## Figures and Tables

**Figure 1 micromachines-13-01533-f001:**
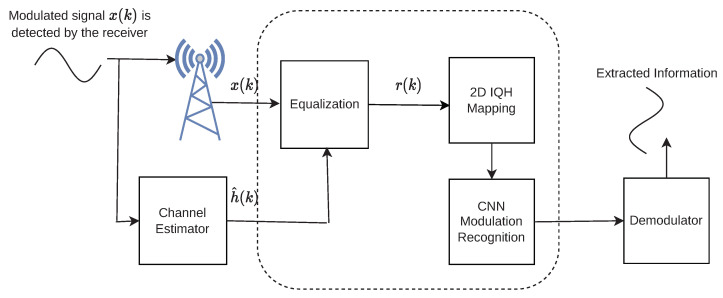
The conceptual diagram of the proposed receiver.

**Figure 2 micromachines-13-01533-f002:**
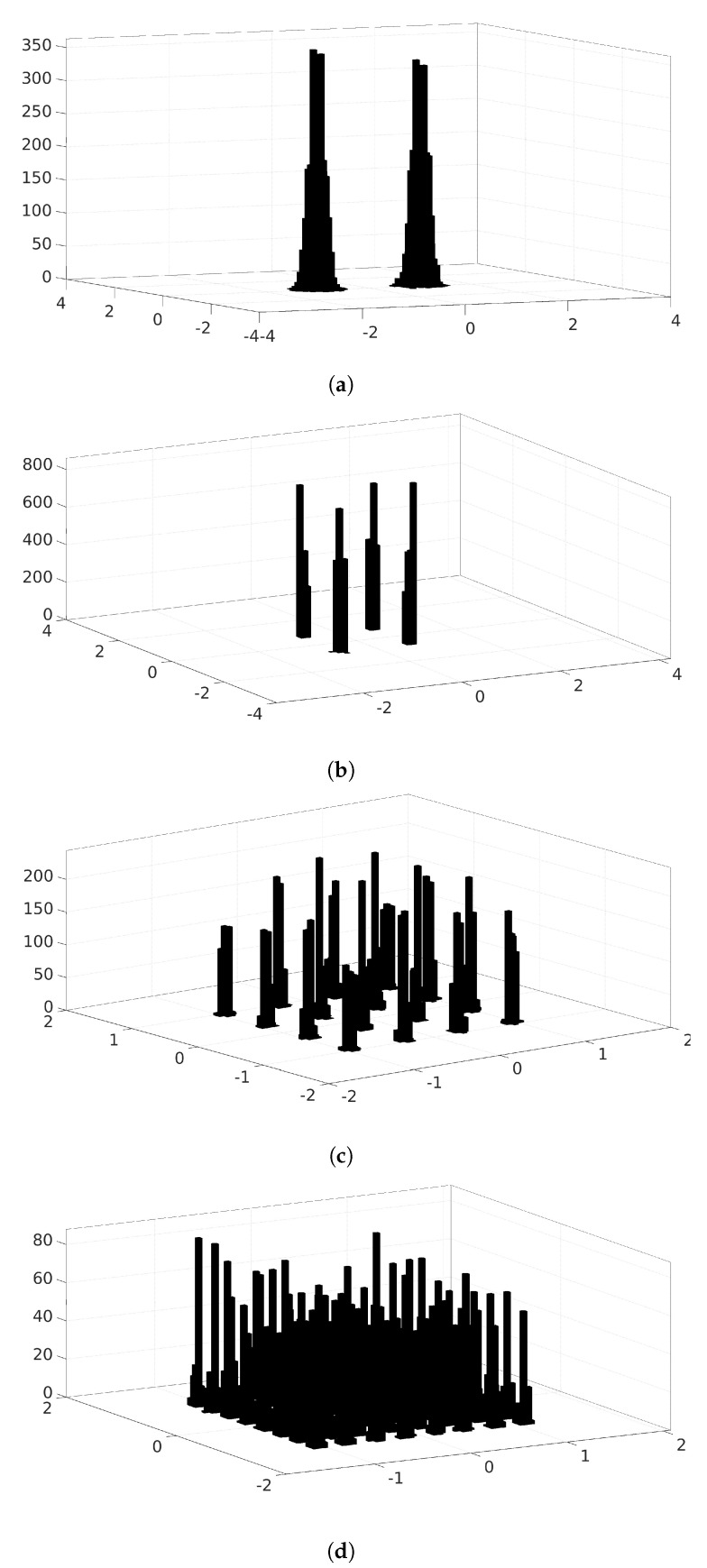
Histogram representation of various modulation schemes. Both the *x* and *y* axes are divided into 80 levels over the intervals x,y∈[−4,4]. (**a**) BPSK scheme histogram representation; (**b**) QPSK scheme histogram representation; (**c**) 16-QAM scheme histogram representation; and (**d**) 64-QAM scheme histogram representation.

**Figure 3 micromachines-13-01533-f003:**
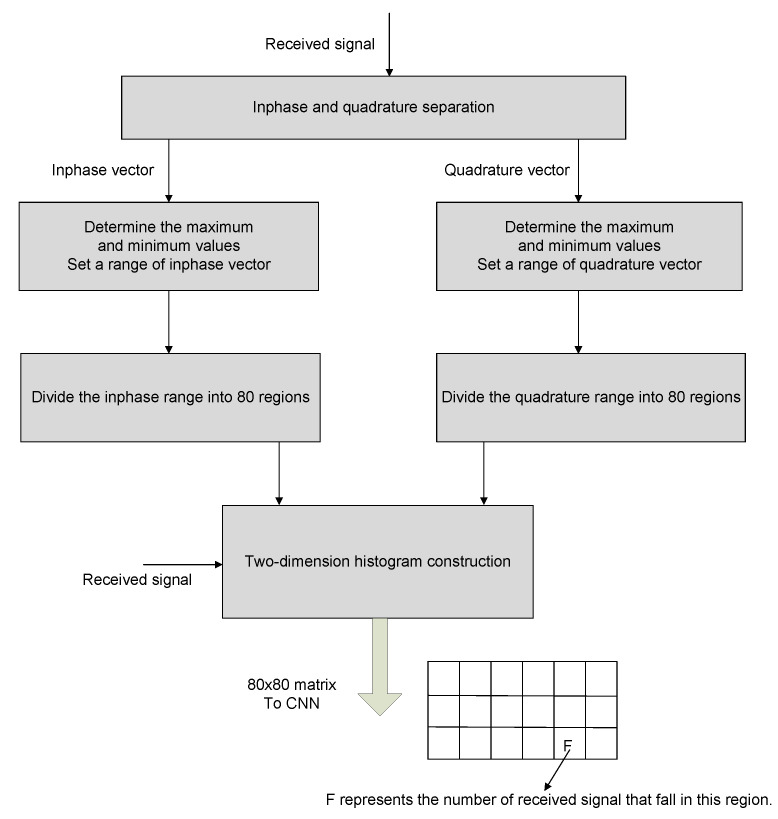
Data generation and feature extraction process.

**Figure 4 micromachines-13-01533-f004:**
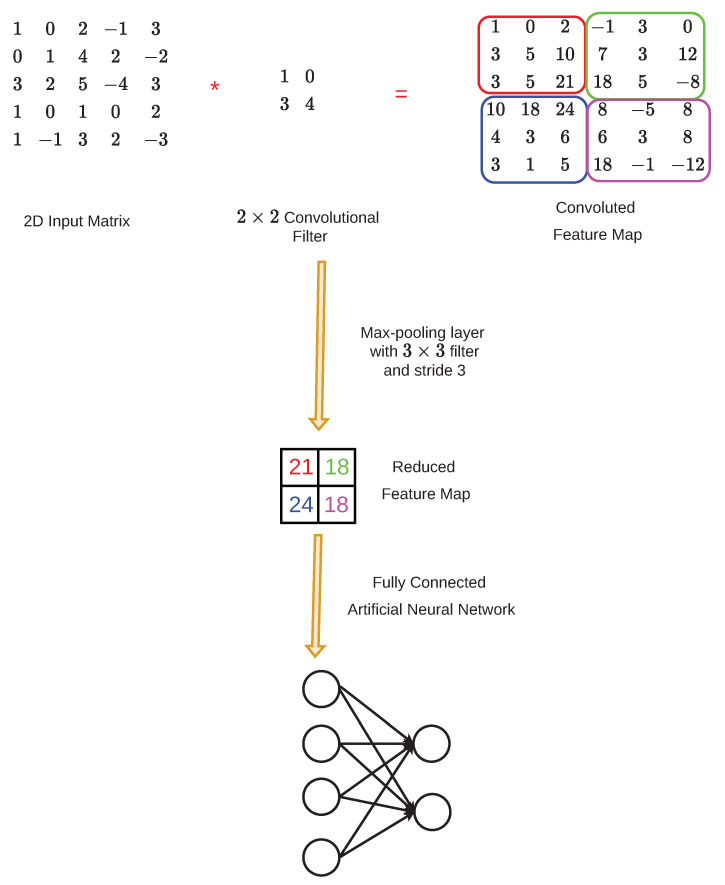
A general illustration of the convolutional layer, the pooling layer, and the fully connected layer in a CNN.

**Figure 5 micromachines-13-01533-f005:**
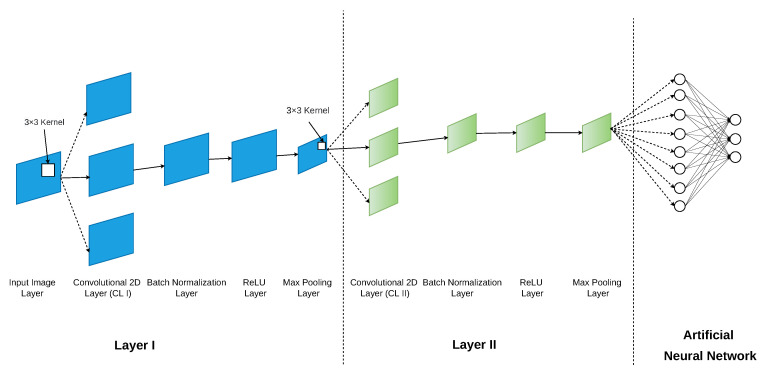
Designed convolutional neural network structure for the proposed algorithm.

**Figure 6 micromachines-13-01533-f006:**
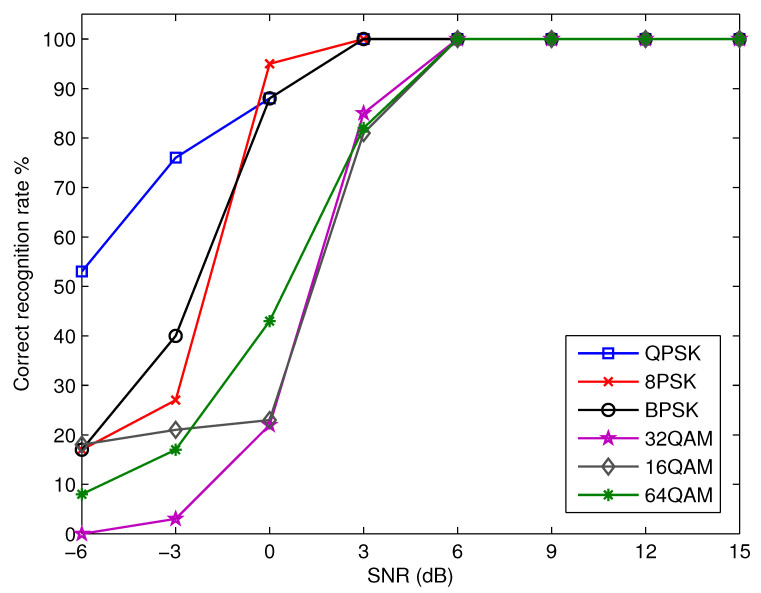
The recognition performance of the proposed algorithm.

**Figure 7 micromachines-13-01533-f007:**
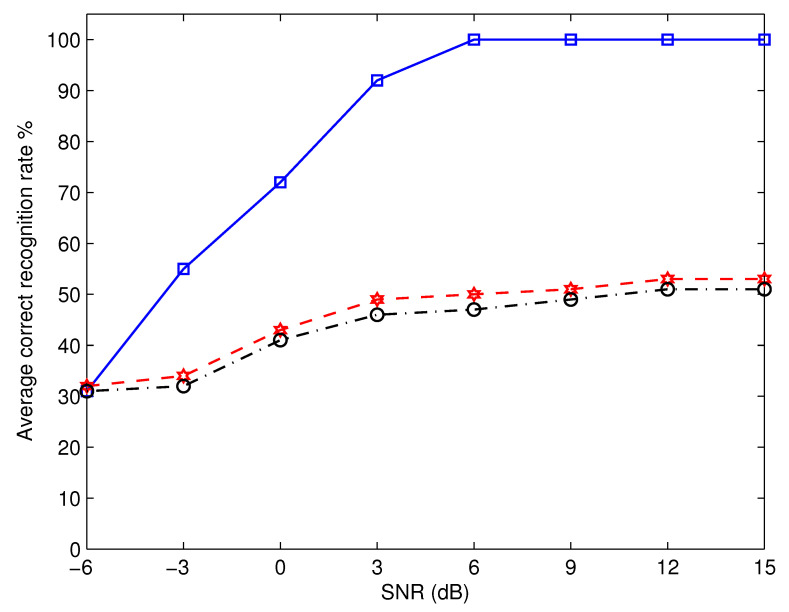
Comparison between the proposed algorithm and that reported in [25]. The proposed algorithm is represented by the square marker, and SVM and NN algorithms of [25] are described by hexagram and circle markers, respectively.

**Figure 8 micromachines-13-01533-f008:**
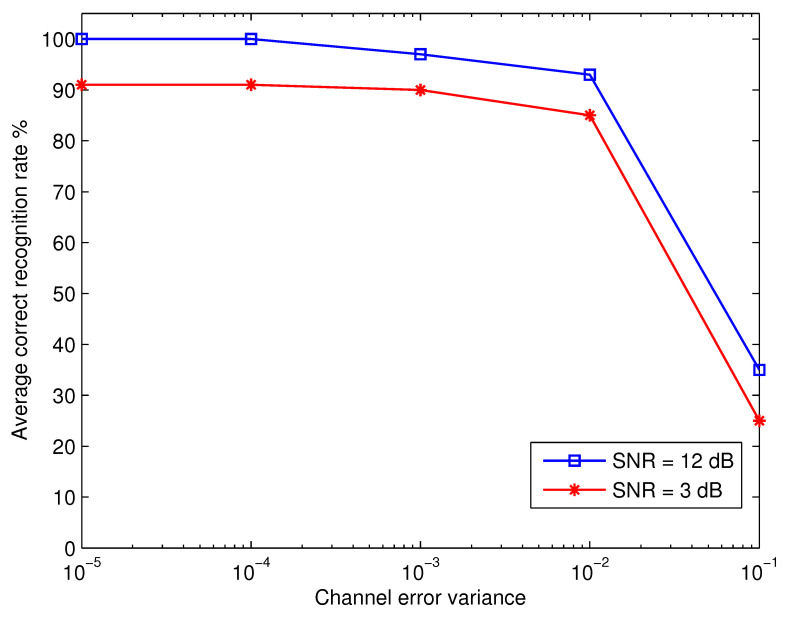
Effect of channel estimation error on the performance of the proposed scheme.

**Figure 9 micromachines-13-01533-f009:**
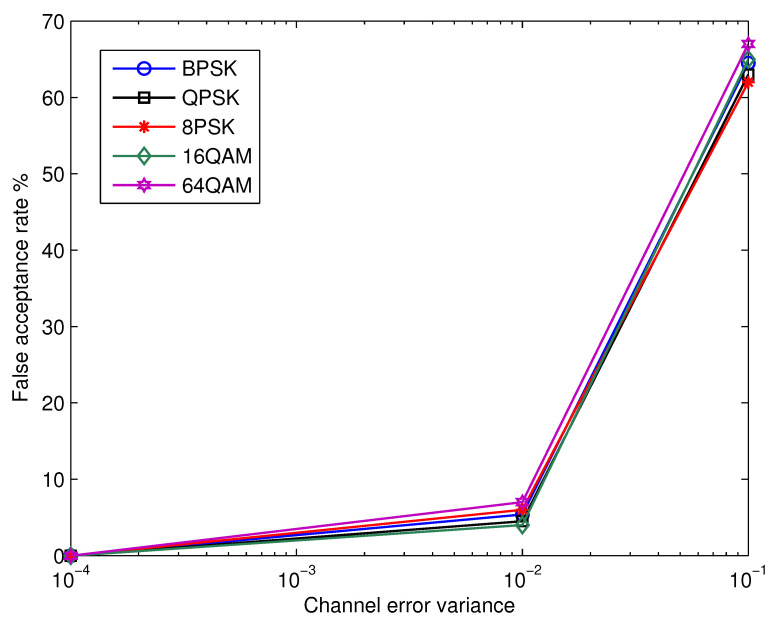
False acceptance rate as a function of the channel error variance at SNR = 12 dB.

**Figure 10 micromachines-13-01533-f010:**
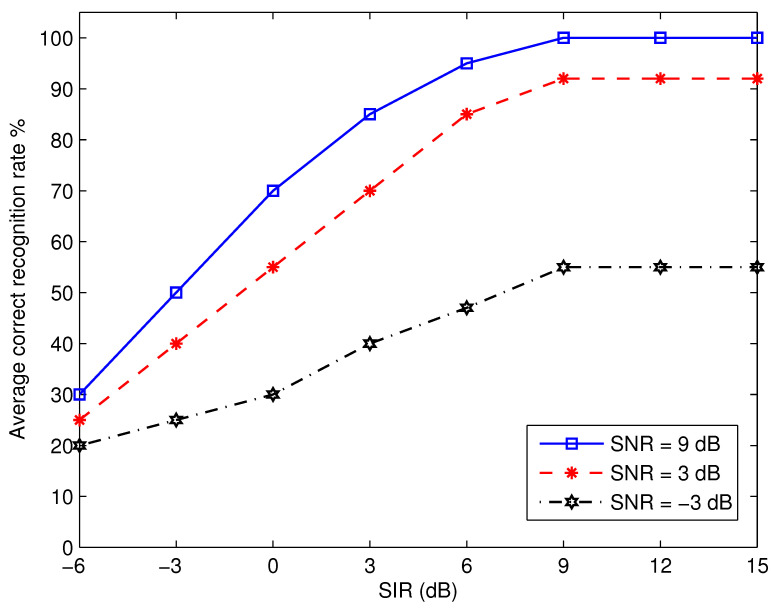
Performance of the proposed algorithm in the presence of narrow-band interference.

**Table 1 micromachines-13-01533-t001:** Data-set generation over different SNR values. A number of 9600 received signals. Note that for each pair of combination we generate 200 signals.

	$r_{1}$ (SNR = −6 dB, BPSK)	$r_{2}$ (SNR = −6 dB, QPSK)	…	$r_{9600}$ (SNR = 15 dB, 64QAM)
8192 Samples			.	
			.	
			.	
			.	
			.	

**Table 2 micromachines-13-01533-t002:** Various parameters of the designed CNN.

CNN Parameter	Type/Value
Optimizer	Adam
Initial Learning Rate	0.001
Decay Rate of Squared Gradient Moving Average	0.99
Max Number of Epochs	10
Mini Batch Size	64
Number of Deep Layers	2
Number of Filters in Each Layer	128

**Table 3 micromachines-13-01533-t003:** Personal computer specifications.

Processor	Intel(R) Core(TM) i7-4790K CPU @ 4.00 GHz
RAM	16 GB
Graphics Card	AMD Radeon (TM) R9 390 Series
SSD	Samsung SSD 870 EVO 500 GB
Operating System	Ubuntu 22.04.1 LTS
Software	Matlab-R2022b

**Table 4 micromachines-13-01533-t004:** Confusion matrix of the proposed algorithm at SNR = 12 dB and $\sigma_{h}^{2}=10^{-4}$.

	BPSK	QPSK	8PSK	8QAM	32QAM	64QAM
BPSK	100	0	0	0	0	0
QPSK	0	100	0	0	0	0
8PSK	0	0	100	0	0	0
8QAM	0	0	0	100	0	0
32QAM	0	0	0	0	100	0
64QAM	0	0	0	0	0	100

**Table 5 micromachines-13-01533-t005:** Confusion matrix of the proposed algorithm at SNR = 12 dB and $\sigma_{h}^{2}=10^{-2}$.

	BPSK	QPSK	8PSK	8QAM	32QAM	64QAM
BPSK	95	1	1	1	1	1
QPSK	1	96	1	1	1	0
8PSK	1	2	94	1	1	1
8QAM	1	1	1	95	1	1
32QAM	1	1	1	2	94	1
64QAM	1	1	1	1	2	94

**Table 6 micromachines-13-01533-t006:** Confusion matrix of the proposed algorithm at SNR = 12 dB and $\sigma_{h}^{2}=10^{-1}$.

	BPSK	QPSK	8PSK	8QAM	32QAM	64QAM
BPSK	38	15	14	11	9	13
QPSK	10	37	16	15	12	10
8PSK	13	19	38	12	10	8
8QAM	7	8	20	36	15	14
32QAM	5	10	7	21	34	23
64QAM	7	8	14	16	22	33

## Data Availability

Not applicable.

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
