# Peer review of "Novel Deep-Learning Modulation Recognition Algorithm Using 2D Histograms over Wireless Communications Channels"

_micromachines, 2022, doi:10.3390/mi13091533_

Round 1

Reviewer 1 Report

1. Motivations of the paper are not clear.

2. Contributions are not mentioned. Most importantly, the structure of the Introduction section is very poor. It should be like:

     A brief discussion about the topic (1st paragraph)

     The problem of the topic (1st or 2nd paragraph)

     Existing solutions of the topic along with the problems of the existing schemes/solutions. (2nd or 3rd paragraph)

     Brief details about the proposed scheme. (3rd or 4th paragraph)

     Contributions (point-wise)

     Structure of the paper (last paragraph)

3. As Related Works section is not given, the Introduction section must be very strong.

4. Rename section 2 as "Preliminary Studies".?

5. There must not be ". (dot) or , (comma)" in many equations.

6. Which data preprocessing technique is used in the proposed scheme and why?

7. Caption of the table must be above the table. 

8. Why deep learning is used in the proposed scheme?

9. The proposed scheme is unstructured. It is hard to identify the novelty of the proposed work.

10. In section 5, add a subsection "Experimental Environment". Then, in another subsection "Results and Discussion", discuss the results. 

11. Figures must be placed at the appropriate location.

12. Technical discussion on results is not given. Moreover, the results are not convincing.

13. The English language is very poor.

14. The organization of the paper is poor.

15. Important references are missing and all the details of the references are not given. Include the following references to strengthen the Reference section.

"SDN-based intrusion detection system for IoT using deep learning classifier (IDSIoT-SDL)", CAAI Transactions on Intelligence Technology, 2021. DOI: https://doi.org/10.1049/cit2.12003 

“Experience replay-based deep reinforcement learning for dialogue management optimisation”, ACM Transactions on Asian and Low-Resource Language Information Processing, 2022. DOI: https://doi.org/10.1145/3539223 

"Network anomaly detection using deep learning techniques", CAAI Transactions on Intelligence Technology, 2022. DOI: https://doi.org/10.1049/cit2.12078

“Multi-authority CP-ABE-based access control model for IoT-enabled healthcare infrastructure”, IEEE Transactions on Industrial Informatics, 2022. DOI: 10.1109/TII.2022.3167842

"Deep learning for time series forecasting: The electric load case", CAAI Transactions on Intelligence Technology, 2022. DOI: https://doi.org/10.1049/cit2.12060

"QEST: Quantized and efficient scene text detector using deep learning", ACM Transactions on Asian and Low-Resource Language Information Processing, 2022. DOI: https://doi.org/10.1145/3526217

“A novel elliptic curve cryptography based system for smart grid communication”, International Journal of Web and Grid Services, vol. 17, no. 4, pp. 321-342, 2021. DOI: 10.1504/ijwgs.2021.10040914

Author Response

Please see the attached response. 

Reviewer 2 Report

The paper address the problem of automatic modulation recognition. The key contribution is related to the application of relatively simple CNN structure in combination of in-phase quadrature histograms to classify the modulation type used for the transmission. The paper however lacks details in many aspects, mainly in description of the algorithm and experimental setup.

Comments:

Introduction

It is recommended to clearly state the contribution of the Authors in a separate paragraph (and using bullets).

What was the motivation for selection of the particular set of modulations for recogmition?

Data Preprocessing

Theh paper would benefit from graphical presentation of the key preprocessing steps (diagram?).

Manual adjustment of number of histogram bins used for classification - it appears that the number of bins was subject of some kind of optimization. What criteria were used to assess the results? Could this process be automated in some way (could more structured optimization approaches be used)?

Proposed MR Algorithm

The papers provides almost no details on the proposed modulation recognition algorithm. According to the Instructions for Authors:
"Materials and Methods: They should be described with sufficient detail to allow others to replicate and build on published results. New methods and protocols should be described in detail while well-established methods can be briefly described and appropriately cited. Give the name and version of any software used and make clear whether computer code used is available. Include any pre-registration codes."

Simulation Work - Results

The Authors provide no implementation details - was the algorithm implemented using any of the reference libraries? Or fully developed and implemented by the researchers?

What computing platform was used for simulations, and what are the computational resources required to run the algorithm (is it considered to create a complete MR system with the use of popular SDR platforms?)? What was the training time, and classification time?

Have confusion matrices been obtained for classification of different modulation types?

How the algorithm would perform in the presence of other modulated interfering signals?

Author Response

Please see the attached response. 

Round 2

Reviewer 1 Report

This revised version can be accepted for publication. 

Author Response

Thank you for your positive opinion. 

Reviewer 2 Report

The Authors extensively addressed the issues raised in the first round of review - thank you.

The last thing to discuss:

- the Authors show the results of the correct recognition rate as a function of various parameters. Have any other classification parameters been analysed and compared, like false rejection rate, false acceptance rate (somewhat related to the presented confusion or matching matrices)?

Author Response

Please find the attached response. Thank you 
